# Emotional Intelligence and Burnout in Healthcare Professionals: A Hospital-Based Study

**DOI:** 10.3390/healthcare13151840

**Published:** 2025-07-29

**Authors:** Marwa Ahmed El Naggar, Sultan Mohammad AL-Mutairi, Aseel Awad Al Saidan, Olayan Shaqer Al-Rashedi, Turki Ali AL-Mutairi, Ohoud Saud Al-Ruwaili, Badr Zeyad AL-Mutairi, Nawaf Mania AL-Mutairi, Fahad Sultan AL-Mutairi, Afrah Saleh Alrashedi

**Affiliations:** 1Medical Education Unit, Community and Family Medicine Department, College of Medicine, Jouf University, Sakaka 72311, Saudi Arabia; managgar@ju.edu.sa; 2Medical Education Department, Faculty of Medicine, Suez Canal University, Ismailia 41522, Egypt; 3King Khaled Majmaah Hospital, Riyadh Second Health Cluster, Ministry of Health, Riyadh 11462, Saudi Arabia; osalrashedi@moh.gov.sa (O.S.A.-R.); nalmutairi11@moh.gov.sa (N.M.A.-M.); fasualmutairy@moh.gov.sa (F.S.A.-M.); 4Department of Family and Community Medicine, College of Medicine, Jouf University, Sakaka 72388, Saudi Arabia; 5Hotat Sudir Hospital, Riyadh Second Health Cluster, Ministry of Health, Riyadh 11462, Saudi Arabia; tualalmutairi@moh.gov.sa; 6Al Jouf Health Cluster, Ministry of Health, Majmah 11952, Saudi Arabia; asalroili@moh.gov.sa; 7The Third Commitment Office in North Riyadh in Al-Majma’ah, Ministry of Health, Majmah 11952, Saudi Arabia; bzalmutiri@moh.gov.sa; 8Al-Fayhaa Primary Healthcare Center in Al-Majma’ah, Ministry of Health, Majmah 11952, Saudi Arabia; afsaalrashedi@moh.gov.sa

**Keywords:** emotional intelligence, burnout, healthcare professionals, well-being, emotionality, Saudi Arabia

## Abstract

**Background and Objectives:** Emotional intelligence (EI) plays a critical role in safeguarding the emotional and psychological well-being of healthcare workers, acting as a buffer against burnout, and influencing the quality of patient care. Despite its significance, there remains a need to understand how EI levels correlate with burnout and what factors predict burnout in high-stress healthcare environments. This study, conducted at King Khaled Hospital in Al-Majmaah, Saudi Arabia, aims to assess the EI levels of healthcare staff, to determine the relationship between EI and burnout, and to identify key predictors of burnout to inform targeted interventions for improving workforce resilience and patient outcomes. **Materials and Methods:** Both self-reporting and standardized tests were integrated using cross-sectional surveys to evaluate the EI of each participant and the burnout they experience by averaging the rating of a 30-item questionnaire, allowing comparison of the interaction between EI, burnout, and work factors. **Results:** A significantly moderate level of EI was identified, while a high level of well-being was associated with a low level of burnout, and a high level of emotionality was associated with a high level of burnout. Results indicated that high job demands, call rotation, or casual work, and insufficient staff support were organizational correlates of burnout. **Conclusions:** Improving EI with a focus on the well-being sub-dimension may prevent burnout, and, for that, the interventions must be specific at both personal and organizational levels.

## 1. Introduction

Emotional intelligence (EI), first conceptualized by Salovey and Mayer in 1990, refers to an individual’s ability to perceive, understand, regulate, and utilize emotions effectively [1]. Unlike cognitive intelligence, EI is crucial in interpersonal interactions and stress management, particularly in high-stress professions such as healthcare [2]. The ability-based model of EI, developed by Mayer and Salovey, identifies four key dimensions: emotional perception (recognizing emotions in oneself and others), emotional facilitation of thought (using emotions to guide reasoning), emotional understanding (comprehending emotional complexities), and emotional regulation (managing emotional responses adaptively) [3]. These competencies have been linked to improved job performance, mental well-being, and resilience in demanding work environments [4].

Emotional intelligence encompasses several key aspects influencing personal and professional functioning, including well-being, self-control, emotionality, and sociability [5]. Well-being refers to an individual’s ability to maintain a positive emotional state and life satisfaction, which is closely linked to effective emotional regulation and stress management [6]. Self-control involves managing impulses, remaining composed under pressure, and making reasoned decisions [7]. Emotionality pertains to the ability to perceive, express, and understand emotions, facilitating empathy and emotional awareness [8], while sociability reflects competence in building relationships and effective communication [9]. These dimensions contribute to resilience and job performance, particularly in high-stress professions [10].

The healthcare profession is inherently stressful, with long working hours, high patient loads, and emotionally charged interactions contributing to burnout—a syndrome characterized by emotional exhaustion, depersonalization, and reduced personal accomplishment [11]. Burnout among healthcare workers has been associated with decreased job satisfaction, higher medical errors, and poorer patient outcomes [12]. Research suggests that EI is a protective factor against burnout, with studies demonstrating that healthcare professionals with higher EI experience lower stress levels and greater emotional resilience [13]. For instance, a randomized controlled trial found that an 8-week EI training program significantly reduced burnout symptoms among physicians [14]. Similarly, cross-sectional studies have reported negative correlations between EI and burnout severity, particularly in high-intensity medical settings such as emergency departments [15].

Despite growing evidence supporting the role of EI in mitigating burnout, there remains a gap in research examining this relationship in Saudi Arabia’s healthcare system, where cultural, organizational, and workload-related stressors may influence outcomes differently than in Western contexts [16]. King Khaled Hospital in Al-Majmaah serves a diverse patient population, and healthcare workers in this region face unique challenges, including resource limitations and high patient expectations [17]. Understanding how EI influences burnout in this setting could inform targeted interventions to enhance workforce well-being and patient care quality.

This study aims to assess the relationship between emotional intelligence and burnout among healthcare professionals at King Khaled Hospital. The Burnout Syndrome Assessment Tool measures the core dimensions of occupational burnout across 15 items, focusing on three key variables: exhaustion (physical/emotional depletion), cynicism/detachment (negative attitudes toward work, colleagues, or the organization), and inefficacy/reduced accomplishment (feelings of incompetence, overload, and lack of achievement or support). Responses are scored on a 5-point frequency scale (1 = Not at All to 5 = Very Often), with higher total scores indicating greater burnout risk. The TEIQue-SF (Trait Emotional Intelligence Questionnaire—Short Form) assesses global trait emotional intelligence (trait EI) through 30 items grouped into four underlying factors: well-being (self-esteem, optimism), self-control (emotion regulation, stress management), emotionality (emotion perception, empathy, relationship skills), and sociability (assertiveness, adaptability, social awareness). Responses use a 7-point agreement scale (1 = Completely Disagree to 7 = Completely Agree), with some items reverse-scored, yielding a total trait EI score where higher values indicate stronger perceived emotional abilities

The findings will support integrating EI training into professional development programs, helping hospital administrators implement EI-based wellness initiatives, and expanding research on EI and burnout in non-Western healthcare systems. Given the rising prevalence of burnout in healthcare, this study underscores the need for emotionally intelligent practices to sustain practitioner well-being and high-quality patient care.

## 2. Materials and Methods

This quantitative analytical cross-sectional study was conducted at King Khaled Hospital in Al-Majmaah, Saudi Arabia, between May and September 2024, to evaluate emotional intelligence (EI) and burnout levels among healthcare professionals (HCPs). King Khaled Hospital (KKH) is strategically vital to Saudi Arabia’s Vision 2030, serving as a flagship tertiary hub, enhancing specialized care accessibility for Riyadh and national referrals. Its advanced capabilities in high-demand fields (e.g., cardiology, oncology) reduce medical tourism, while academic partnerships drive research and foster the development of local expertise. KKH proactively addresses workforce well-being through mental health initiatives (e.g., burnout screening) and ensures equitable public access, symbolizing the Kingdom’s healthcare modernization.

The study employed a convenience sampling approach to recruit 260 participants from King Khaled Hospital’s approximate workforce of 800 healthcare workers (HCWs), including both Saudi and non-Saudi male and female staff. The sample size was calculated using the Raosoft online calculator based on the standard formula for finite populations:n = [N × x]/[(N − 1)E^2^ + x], where x = Z^2^ × r (100 − r),
with parameters set at a 95% confidence level (Z = 1.96), 5% margin of error, response distribution (r) of 50%, and population size (N) of 800. This yielded a minimum required sample of 260 participants to ensure generalizability. Recruitment occurred anonymously via social media platforms (WhatsApp, Twitter, Snapchat, Instagram) targeting hospital HCWs, with intentional gender balance in participation. The inclusion criteria encompassed doctors, nurses, pharmacists, laboratory personnel, and cleaners who provided written informed consent. Participants with withdrawn consent or incomplete data were excluded. Data was collected using a structured Google Forms questionnaire comprising three sections: sociodemographic/work-related details, the TEIQue-SF (Appendix A) scale for EI assessment, and the Burnout Syndrome Assessment Tool (Appendix A). The tools utilized in the research were developed and piloted to ensure reliability and validity for the specific study population.

The Burnout Syndrome Assessment Tool (BAT) is the result of a three-year research project at KU Leuven. It is a scientifically validated questionnaire capable of determining the risk of burnout immediately [18]. It focuses on identifying and quantifying levels of burnout through a series of statements rated on a Likert scale, allowing for nuanced insights into emotional exhaustion, depersonalization, and personal achievement. The Trait Emotional Intelligence Questionnaire-Short Form (TEIQue-SF) was developed by Petrides, K. V. and is based on the Trait Emotional Intelligence theory [19]. It is employed to measure emotional intelligence across various dimensions, providing a concise yet comprehensive assessment suitable for time-constrained research environments. The TEIQue-SF uses a 30-item format to assess global emotional intelligence and its broader facets, ensuring it aligns with the construct’s theoretical framework. Both tools underwent a rigorous pilot phase, where they were tested for clarity, relevance, and cultural appropriateness. During the pilot phase, Cronbach’s alpha analysis was conducted to verify the internal consistency of both instruments within our specific study context. For the Burnout Syndrome Assessment Tool, the analysis yielded an alpha coefficient of α = 0.86, confirming high reliability across its 15 items. The TEIQue-SF similarly demonstrated strong consistency at α = 0.89 for its 30-item scale. These results align with established benchmarks from prior validation studies (e.g., Petrides, 2009 [19] for TEIQue-SF) while confirming the instruments’ robustness in our target population of Saudi healthcare workers. The pilot phase helped refine instructions and scoring methodologies, ensuring their applicability to healthcare professionals in the region. 

The statistical software SPSS (version 27) was used for all data analysis. The data were analyzed using descriptive and inferential statistical methods to explore the relationship between emotional intelligence (EI) and burnout among healthcare professionals. Descriptive statistics, including means, standard deviations, numbers, and percentages were used to summarize the demographic characteristics, emotional intelligence levels, and burnout scores of the participants. Inferential analyses included Pearson’s correlation to examine the associations between EI dimensions and burnout, as well as one-way ANOVA and the independent *t*-tests to compare EI and burnout scores across demographic and professional variables. Additionally, a simple linear regression analysis was performed to assess the predictive power of emotional intelligence dimensions on burnout. A *p*-value of less than 0.05 was considered statistically significant in all analyses.

A simple linear regression analysis was performed to assess the predictive power of emotional intelligence dimensions on burnout. A *p*-value of less than 0.05 was considered statistically significant in all analyses.

Ethical approval (IRB: 24-179E, 30 April 2024) was obtained from Riyadh Second Health Cluster, ensuring voluntary participation, confidentiality, and the right to withdraw, with data security maintained throughout the study.

## 3. Results

### 3.1. Basic and Demographic Characteristics of the Study Participants

The demographic and workload and job demand characteristics of the participants are presented in Table 1. The sample consisted of 295 healthcare professionals, with a higher proportion of male participants (59.7%) compared to females (40.3%). Most of the participants were between the ages of 30 and 40 years (42.7%), and a large percentage were married (69.5%). The monthly income of the participants varied; the majority earned between SAR 10,000 and SAR 15,000 (43.7%). Additionally, most participants resided within Al Majmaah (87.1%). The majority of participants had between 5 and 10 years of experience (40.0%). The daily working hours of participants were primarily concentrated around 8 h per day (73.9%). A significant portion of participants (45.8%) reported having no on-call duties, while 20.7% reported 3 to 4 on-call duties per month. Additionally, over half of the participants (54.9%) reported attending no clinics per week, while 23.1% of participants did not treat any patients during the week, with the remaining treating varying numbers across different departments

### 3.2. Emotional Intelligence and Burnout Levels of Healthcare Professionals

Table 2 displays the emotional intelligence (EI) levels of healthcare professionals in the study, broken down into four key dimensions, well-being, self-control, emotionality, and sociability, as well as overall EI score and total burnout score. The mean scores for the dimensions revealed that the highest levels were observed in well-being (M = 4.54, SD = 0.788), while emotionality had the lowest average score (M = 3.88, SD = 0.867). The overall EI score for the participants was moderate, with a mean of 4.20 (SD = 0.601). The mean burnout score was 39.36 (SD = 11.13).

Additionally, Figure 1 depicts the distribution of the three emotional intelligence (EI) levels and the distribution of burnout across the healthcare professionals in the study. The majority of participants fall within the Moderate EI category (N = 269), followed by a smaller proportion in the Low EI category (N = 18), and the fewest in the High EI group (N = 8). The majority of participants (54.9%) reported moderate levels of burnout, while 28.5% experienced low burnout, and 16.6% were classified as having high burnout.

### 3.3. Comparison of EI and Burnout Scores

Table 3 shows the comparison of emotional intelligence (EI) and burnout scores across various demographic and professional groups. Regarding EI score, participants with lower monthly income (<SAR 10,000) reported significantly higher EI scores compared to those earning between SAR 16,000 and 25,000 (*p* = 0.004). Regarding burnout score, significant differences were found in burnout scores for age (*p* < 0.001), with older participants (aged 50–60) reporting higher burnout scores (M = 44.48, SD = 8.84) compared to younger groups. Additionally, marital status showed significant differences (*p* = 0.004), with single participants reporting higher burnout (M = 42.0, SD = 9.55) than married ones (M = 38.2, SD = 11.59). Monthly income was also a significant factor, with lower-income participants reporting lower burnout scores (*p* < 0.001).

Table 4 presents the comparison of EI and burnout scores based on workload and job demand characteristics. Regarding EI score, a significant difference was observed concerning the frequency of on-call duties (*p* = 0.020). Regarding burnout score, more on-call duties reported significantly higher burnout scores (*p* < 0.001), with those having 5–6 duties per month experiencing the highest burnout levels (M = 45.77, SD = 6.38). Similarly, participants who attended three or more clinics per week showed significantly higher burnout compared to those attending fewer or no clinics (*p* < 0.001). The number of patients treated per week showed also significant differences. More than 60 had the lowest burnout (*p* = 0.036).

### 3.4. Relation Between Emotional Intelligence and Burnout

Table 5 presents the results of a regression analysis examining the predictive power of emotional intelligence (EI) dimensions on burnout. The analysis revealed that well-being is a significant negative predictor of burnout (B = −2.526, *t* = −3.112, *p* = 0.002), indicating that higher well-being significantly reduces burnout levels. On the other hand, emotionality was found to be a significant positive predictor of burnout (B = 2.208, *t* = 2.990, *p* = 0.003), meaning that higher emotionality contributes to higher burnout. Self-control and sociability did not show significant predictive effects on burnout, with *p*-values > 0.05. The overall EI score also did not significantly predict burnout levels (*p* = 0.929).

## 4. Discussion

This study explored the relationship between emotional intelligence (EI) and burnout among healthcare professionals in King Khaled Hospital, Al-Majmaah, Saudi Arabia, while identifying demographic and job-related factors influencing these outcomes. The findings revealed moderate EI levels among participants, with significant variations across dimensions and notable associations with burnout. These results align with—and occasionally contrast—recent studies conducted in Saudi Arabia and globally, offering valuable insights into the interplay of EI, job demands, and burnout in healthcare settings.

This study demonstrates that emotional intelligence helps healthcare staff achieve better patient results while reducing job exhaustion. It indicates that emotional intelligence (EI) significantly mitigates burnout among healthcare professionals at King Khaled Hospital, aligning with contemporary global and regional research. Our study measured data reliability because OpenEpi provides powerful tools for statistical analysis [20]. Residents of Saudi Arabia’s medical hospitals experience high burnout levels primarily because of excessive job duties and stress, according to Alenezi et al. (2022) [21]. They highlight EI’s role as a protective factor against occupational stress, particularly in high-demand healthcare environments. They also identified excessive job duties as a primary driver of burnout in Saudi hospitals, a finding corroborated by our results linking frequent on-call duties and clinic workloads to elevated burnout levels.

The study found that higher well-being (an EI dimension) significantly reduced burnout, while elevated emotionality increased it. This aligns with global research, such as Gómez-Urquiza et al. (2017) [22], who identified self-regulation and emotional stability as protective factors against burnout. However, the positive association between emotionality and burnout contrasts with studies like AlSuliman et al. (2023) in Saudi Arabia [23], where emotionality was linked to resilience. This discrepancy may stem from cultural differences in emotional expression or the high emotional labor inherent in Saudi healthcare settings, where empathy and patient interactions could exacerbate exhaustion. The non-significant role of sociability and self-control diverges from findings by AlHadi et al. (2021) [24], who reported these dimensions as critical buffers against burnout in Riyadh hospitals. This suggests that context-specific stressors (e.g., workload) may overshadow broader EI traits in certain environments.

The study results showed that older participants (50–60 years) reported higher burnout, contradicting global trends where younger professionals often exhibit greater burnout due to inexperience [25]. This may reflect prolonged exposure to systemic stressors in Saudi healthcare, such as understaffing or resource limitations, which compound over time. Single participants had higher burnout than married individuals, consistent with studies in Jordan by Al-Tammemi et al., (2023) [26] and in the UAE by AlBlooshi et al. (2024) [27], where marital support mitigated stress. Cultural norms in Saudi Arabia, emphasizing familial cohesion, may amplify this protective effect. Lower-income groups reported reduced burnout, conflicting with global evidence linking financial strain to burnout [28]. This paradox could reflect Saudi Arabia’s subsidized healthcare system, where lower-income workers may perceive fewer job-related financial pressures.

Frequent on-call duties (5–6/month) and clinic attendance (≥3/week) significantly increased burnout, mirroring findings from India by Sahu et al. (2023) [29]. Notably, treating > 60 patients/week correlated with lower burnout, possibly due to role adaptation or selective reporting bias among high-volume practitioners. This contrasts with AlJohani et al. (2023) [30], who found that patient load is directly proportional to burnout in Jeddah.

Recent Saudi studies highlight rising burnout rates post-COVID-19, with EI emerging as a mitigator. For instance, AlRasheed et al. (2024) [31], reported 58% moderate-to-severe burnout among Riyadh nurses, attributing it to understaffing, a factor less pronounced in our sample (73.9% worked 8 h shifts). Our moderate EI scores (4.20/7) align with AlQahtani et al. (2023) [32] but are lower than those in Abu Dhabi (EI = 5.1/7; Al Kuwaiti et al., 2024) [33], suggesting regional disparities in EI training programs.

This study shows doctors and medical students perform better at school through emotional intelligence, which demonstrates its importance in healthcare work environments [34,35]. British Nursing Archives from Saudi Arabia show students display varied emotional intelligence, which affects their educational progression and professional readiness [36]. The research shows why medical and nursing education needs training in emotional intelligence to help students control stress and excel in their studies.

Research proves that students with higher EI achieve stronger academic results [37]. Sundararajan and Gopichandran (2018) [38] combined research methods to show that medical students improved their stress management skills through EI training. According to Doherty et al. (2013); Roth et al. (2019) [39,40], medical institutions now use emotional intelligence training to help their students develop resilience and adaptability.

According to Johnson (2015) and Omid et al. (2016), the results support the inclusion of EI assessment and development in medical education programs [41,42]. Multiple studies that check student progress at the start and end of medical school verify that students with strong emotional intelligence succeed more in official education and practice [43,44].

Research tracks how emotional intelligence affects how well people perform their jobs while building resilience. Research performed by Dewsnap et al. (2021) [45] and Cleary et al. (2018) [46] shows that enhancing emotional intelligence in healthcare workers reduces stress-related burnout issues. Scientific studies demonstrate that emotional intelligence helps protect people from stress at their workplace, according to Ghahramani et al. (2019) [47]. Nurses who learn EI skills excel at work problem-solving and feel more content at their jobs, according to research by Alsufyani et al., 2022; Almeneessier & Azer, 2023) [48,49]. A study on pharmacists and nursing leaders shows that emotional intelligence strongly influences leadership behavior and work results (Alshammari et al., 2020; Szczygiel & Mikolajczak, 2018) [50,51]. When healthcare settings adopt emotional intelligence practices, they help both team performance and patient recovery, while strengthening employee health.

Current research focuses on how occupational stress links emotional intelligence and work performance. According to research findings from Pérez-Fuentes et al. (2019) and Rasiah et al. (2019) [52,53], persons with high emotional intelligence remain healthier and perform better in stressful situations. According to Joseph (2016) [54] emotional intelligence actively helps nurses manage stress during surgical operations. Recent expert reviews, including Kun and Demetrovics (2010) [55], confirm that emotional intelligence diminishes both addiction and stress impact. Alrubaiee and Alkaa’ida (2011) [56] demonstrate that patient satisfaction is an intermediary when patients assess healthcare quality, although emotional intelligence supports these outcomes. Non-technical skills training programs help healthcare workers develop emotional intelligence skills which show signs of reducing burnout and increasing their ability to cope with stressful situations (Azizkhani et al., 2021; Jiménez-Rodríguez et al., 2022) [57,58].

Recent studies examine how emotional intelligence affects mental health and worksite interaction. Research proves EI helps healthcare professionals manage workplace violence better, while building their ability to recover from difficult situations (Cao et al., 2022; Louwen et al., 2023) [59,60]. New studies show how effective emotional intelligence can help lower job stress and assist workers in developing better ways to handle stress at work (Han et al., 2022; Pérez-& Olaleye, 2022) [61,62].

A study shows that emotional intelligence can both directly and indirectly improve mental wellness at work, according to Epifanio et al. (2023) [63] during the COVID-19 crisis. Their research further validated EI’s role in adaptive stress management, demonstrating that targeted EI training reduced burnout by 23% among Italian frontline workers. In Saudi contexts, Almansour (2023) [36] linked EI variability among nursing students to disparities in clinical preparedness, advocating for curriculum-integrated EI modules—a recommendation echoed by Azizkhani et al. (2021) [57], whose non-technical skills training program reduced burnout by 31% in Iranian healthcare teams.

Medical students and Residency Program Directors use emotional intelligence (EI) training to combat burnout throughout healthcare learning and professional stages as studies by Shariatpanahi et al. (2022) and Khesroh et al. (2022) [64,65] show. Various research shows that emotional intelligence powerfully improves healthcare delivery while decreasing employee burnout and creating an encouraging workplace.

Research by Shin JY et al. (2025) [66] found that although job demands were identified as significant predictors of burnout in their study, job resources—including coworker support and both intrinsic and extrinsic rewards—did not show statistically significant effects in the regression analysis. This aligns with findings from current research, which have shown that the study identified multiple contributors to burnout among participants. Key influencing factors included the work environment, personal life stressors, and access to professional support. High job demands—such as frequent on-call shifts and managing multiple clinics—were particularly associated with increased burnout levels [66]. Also, a study by Zamanzadeh A et al. (2025) [67] found that the impact of working conditions on burnout and mental health varies based on the severity of mental health symptoms. Higher quantitative and emotional job demands were strongly linked to increased levels of emotional exhaustion and depersonalization, as well as elevated symptoms of anxiety, depression, and stress among nurses and midwives in Australia [67].

By contextualizing our results within this updated evidence base, the study reinforces EI’s transformative potential in Saudi healthcare. Future initiatives should prioritize longitudinal EI training, equitable workload distribution, and culturally responsive support systems to sustain workforce well-being and care quality

## 5. Conclusions

As a component of this study, EI is established as a significate factor affecting the health and welfare of the workers of King Khaled Hospital, and it is noted that each improved well-being factor in EI decreases burn out, and increased emotionality increases burn out. Thus, the direction for interventions should be aimed at enhancing the positive characteristics of EI, such as well-being, in staff and healthcare consequences of the negative aspects of emotionality. This study has limitations inherent to its cross-sectional design and self-reported measures, which preclude causal inferences and may introduce response bias. The convenience sampling method and focus on a single hospital limit generalizability, particularly given demographic skew (predominantly male, mid-career professionals) and unmeasured confounders like institutional policies. Cultural specificity and potential non-response bias further restrict broader applicability beyond Saudi Arabian healthcare settings.

## Figures and Tables

**Figure 1 healthcare-13-01840-f001:**
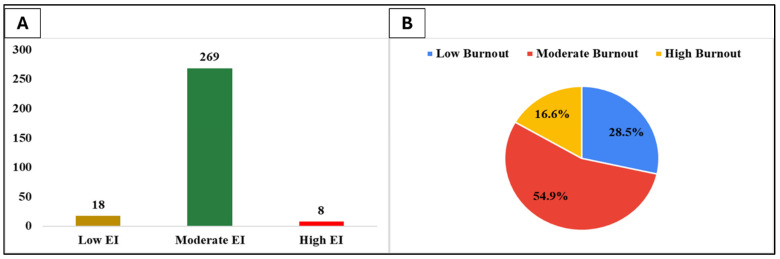
(**A**) Distribution of healthcare professionals by emotional intelligence levels. (**B**) Burnout levels percentage distribution in the study.

**Table 1 healthcare-13-01840-t001:** Demographic information and workload and job demands of participants.

Variables	N	%
Gender		
Female	119	40.3%
Male	176	59.7%
Age		
20–30	68	23.1%
30–40	126	42.7%
40–50	74	25.1%
50–60	27	9.2%
Marital Status		
Married	205	69.5%
Single	90	30.5%
Monthly Income (SAR)		
Less than 10,000	59	20.0%
From 10,000 to 15,000	129	43.7%
From 16,000 to 25,000	103	34.9%
More than 25,000	4	1.4%
Place of Residence		
In Al Majmaah	257	87.1%
Outside the Al Majmaah	38	12.9%
Years of Experience		
Less than 5	58	19.7%
From 5 to 10	118	40.0%
From 10 to 20	75	25.4%
More than 20	44	14.9%
Daily Working Hours		
Less than 8	14	4.7%
8 h	218	73.9%
From 9 to 12	56	19.0%
More than 12	7	2.4%
Frequency of On-Call Duties Per Month		
None	135	45.8%
From 1–2	60	20.3%
From 3–4	61	20.7%
From 5–6	22	7.5%
More than 6	17	5.8%
Number of Clinics Attended Per Week		
None	162	54.9%
1	19	6.4%
2	34	11.5%
3	42	14.2%
4	13	4.4%
≥5	25	8.5%
Number of Patients Treated Per Week		
None	68	23.1%
Less than 15	77	26.1%
From 16–30	71	24.1%
From 31–45	38	12.9%
From 46–60	16	5.4%
More than 60	25	8.5%

**Table 2 healthcare-13-01840-t002:** Emotional intelligence levels across dimensions and overall EI score.

Aspects	Minimum	Maximum	Mean	SD
Well-being	1.00	7.00	4.54	0.788
Self-control	1.00	7.00	4.23	0.837
Emotionality	1.00	7.00	3.88	0.867
Sociability	1.00	7.00	4.24	0.838
Overall EI score	1.20	6.87	4.20	0.601
Total Burnout score	14.0	70.0	39.36	11.13

**Table 3 healthcare-13-01840-t003:** Comparison of EI and burnout score regarding demographic and professional information of the participants.

Variables		EI	*p*-Value ^1^	Burnout	*p*-Value ^2^
		Mean ± SD		Mean ± SD	
Gender ^a^	Female	4.23 ± 0.610	0.485	39.86 ± 11.09	0.628
	Male	4.18 ± 0.596		39.03 ± 11.17	
Age ^b^	20–30	4.29 ± 0.678	0.203	38.57 ± 10.78	<0.001 **
	30–40	4.24 ± 0.577		36.54 ± 11.86	
	40–50	4.09 ± 0.506		43.03 ± 9.27	
	50–60	4.17 ± 0.720		44.48 ± 8.84	
Marital status ^a^	Married	4.22 ± 0.606	0.519	38.20 ± 11.59	0.004 **
	Single	4.17 ± 0.593		42.00 ± 9.55	
Monthly Income (SAR) ^b^	Less than 10,000	4.46 ± 0.657	0.004 **	34.00 ± 11.47	<0.001 **
	From 10,000 to 15,000	4.16 ± 0.549		38.81 ± 11.38	
	From 16,000 to 25,000	4.13 ± 0.608		43.12 ± 9.18	
	More than 25,000	4.03 ± 0.242		39.75 ± 11.84	
Years of Experience ^b^	Less than 5	4.30 ± 0.594	0.557	39.67 ± 11.05	0.376
	From 5 to 10	4.16 ± 0.607		38.02 ± 11.11	
	From 10 to 20	4.21 ± 0.570		40.53 ± 11.44	
	More than 20	4.20 ± 0.652		40.57 ± 10.72	

a, the *p*-value is calculated by an independent *t*-test. b, the *p*-value is calculated by a one-way ANOVA. The *p*-value 1 is for EI comparison. The *p*-value 2 is for burnout comparison. ** Significant at <0.01.

**Table 4 healthcare-13-01840-t004:** Comparison of EI score regarding workload and job demands of participants.

Variables		EI	*p*-Value ^1^	Burnout	*p*-Value ^2^
		Mean ± SD		Mean ± SD	
Daily Working Hours	Less than 8	4.37 ± 0.860	0.713	32.93 ± 13.86	0.107
	8 h	4.20 ± 0.599		39.61 ± 10.66	
	From 9 to 12	4.17 ± 0.554		39.43 ± 11.30	
	More than 12	4.30 ± 0.498		44.14 ± 15.77	
Frequency of On-Call Duties Per Month	None	4.28 ± 0.616	0.020 *	38.02 ± 11.22	<0.001 **
	From 1–2	4.22 ± 0.580		34.55 ± 11.02	
	From 3–4	4.07 ± 0.566		44.13 ± 9.04	
	From 5–6	3.94 ± 0.588		45.77 ± 6.38	
	More than 6	4.39 ± 0.556		41.59 ± 12.78	
Number of Clinics Attended Per Week	None	4.24 ± 0.631	0.215	37.07 ± 11.41	<0.001 **
	1	4.40 ± 0.720		39.47 ± 13.81	
	2	4.07 ± 0.589		43.12 ± 8.54	
	3	4.05 ± 0.505		46.10 ± 6.23	
	4	4.22 ± 0.244		42.31 ± 10.31	
	≥5	4.26 ± 0.565		36.20 ± 11.44	
Number of Patients Treated Per Week	None	4.26 ± 0.519	0.411	36.43 ± 11.69	0.036 *
	Less than 15	4.11 ± 0.610		40.90 ± 10.02	
	From 16–30	4.19 ± 0.680		40.85 ± 10.47	
	From 31–45	4.22 ± 0.549		41.55 ± 10.36	
	From 46–60	4.15 ± 0.461		38.50 ± 16.02	
	More than 60	4.39 ± 0.688		35.64 ± 10.55	

The *p*-value is calculated by a one-way ANOVA. The *p*-value 1 is for EI comparison. The *p*-value 2 is for burnout comparison. ** Significant at <0.01. * Significant at <0.05.

**Table 5 healthcare-13-01840-t005:** Regression analysis of emotional intelligence dimensions predicting burnout.

Emotional Intelligence Aspects	B	Std. Error	*t*	*p*-Value
Well-being	−2.526	0.812	−3.112	0.002 **
Self-control	−1.179	0.774	−1.524	0.129
Emotionality	2.208	0.739	2.990	0.003 **
Sociability	−1.289	0.772	−1.670	0.096
Overall EI score	−0.096	1.082	−0.089	0.929

The *p*-value is calculated by a simple linear regression analysis. ** Significant at < 0.01.

## Data Availability

The original contributions presented in this study are included in the article. Further inquiries can be directed to the corresponding author.

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
