# Peer review of "Emotional Intelligence and Burnout in Healthcare Professionals: A Hospital-Based Study"

_healthcare, 2025, doi:10.3390/healthcare13151840_

Round 1
Reviewer 1 Report
Comments and Suggestions for Authors
The manuscript discusses an interesting topic which can be considered current. I want to suggest a few improvements for the manuscript.
Introduction
- Please add 1 paragraph to summarise the variables in the questionnaire.
Materials and methods
- Please justify the selection of King Khaled Hospital.
- Please elaborate further on the sampling design (if I am not mistaken, the population is approximately 800 to 260 participants).
- Which sampling technique was used for the selection of the participants?
- During the pilot phase, was any analysis run to support the statement (Cronbach's alpha)?
Discussion
Fundamentally, your discussion is acceptable. However, a few enhancements can be made to improve the quality of the discussions.
- The interpretation can be done more deeply by relating to available theories and frameworks.
- Results should also be discussed with the underlying fundamentals of possible cultural influences, occupational structures in Saudi healthcare, as well as differences in study instruments.
- Most importantly, the discussion fails to highlight the novel contributions of the study.
Conclusions
- The study limitations in the previous paragraph should be in the conclusion section.
- Please refer back to the objectives when deriving the conclusions.
Author Response
Thank you for review the manuscript. Please see the attached file.

Reviewer 2 Report
Comments and Suggestions for Authors
Thank you for allowing me to review your article. The topics are very scientifically sound throughout. The introduction provides sufficient background and includes all relevant references. The research design was appropriate.
The methods are adequately described, with minor improvements that could be made as follows:
The method for calculating composite scores for emotional intelligence and burnout is briefly mentioned but could be expanded (e.g., how the Likert scale responses are aggregated).
The results are clearly presented: There are minor considerations that could improve your results. Confidence intervals are not provided; adding them would improve the accuracy of the estimates.
Effect sizes (such as R² for regression) and F values ​​are not mentioned, but they would provide understanding of the strength of the associations.
Figure 1 could be improved with better labels or annotations for clarity.
The annotations or axis labels for the figure are briefly mentioned; descriptive labels (such as numerical values ​​or percentages on the columns) could improve clarity.
The conclusion is very brief: it should be expanded based on the findings, and relevant recommendations should be provided.
Author Response

(The authors gave the same response as above.)

Reviewer 3 Report
Comments and Suggestions for Authors
I reviewed the manuscript titled “The Impact of Emotional Intelligence on the Burnout of Healthcare Professionals Working at King Khaled Hospital in Al-Majmaah City, Kingdom of Saudi Arabia.” This study investigates the relationship between emotional intelligence (EI) and the psychological well-being of healthcare workers. The research measures burnout among these workers using a 30-item questionnaire. The findings indicate that high job demands, call rotation or casual work, and insufficient staff support are organizational factors associated with burnout. The paper introduces an important topic and suggests significant policy implications for health authorities, highlighting that improving EI can help prevent burnout. I appreciate this paper because it is well-written and engaging. However, I have a few suggestions for improvements:
- The title is overly wordy. I recommend shortening it to a maximum of 12 words.
- Several recent studies published in 2025 examine the impact of work demands on burnout, but the authors did not cite them. Some of these useful research papers are provided below:
Shin, J. Y., Lee, S. E., & Morse, B. L. Understanding Burnout in School Nurses: The Role of Job Demands, Resources, and Positive Psychological Capital. The Journal of School Nursing, 10598405251342532.
Zamanzadeh, A., Eckert, M., Corsini, N., Adelson, P., & Sharplin, G. (2025). Mental health of Australian frontline nurses during the COVID-19 pandemic: Results of a large national survey. Health Policy, 151, 105214.
- Figure 1 is not very informative as it does not show the percentage of individuals with low emotional intelligence (EI) who experience burnout. I recommend adding an additional graph that illustrates the prevalence of burnout corresponding to each category of EI.
- Additionally, the conclusion is too brief. It lacks both policy recommendations and suggestions for future research extensions.
Addressing these comments will significantly improve the quality of the paper and enhance its readability.
Author Response

(The authors gave the same response as above.)
